# A HAZOP-based hazard identification model for urban gas accidents: Development and empirical validation

Bin Tian[1]*, Haibing Li[1], Xiaojun Cui[1], Zhihong Hu[1], Zibo Zhou[2], Wei Shi[2]

1 College of Modern Science and Technology, China Jiliang University, Yiwu, China, 2 Yiwu Natural Gas Co., Ltd, Yiwu, China

* tianbin20122013@163.com

## Abstract

Urban gas accidents pose significant threats to public safety and urban infrastructure, with traditional hazard identification methods often relying on manual inspections and experience-based judgments, leading to incomplete or inconsistent results. To address these issues, this study proposes a structured hazard identification model based on Hazard and Operability Analysis (HAZOP) deviation theory for urban gas systems. By integrating four key dimensions—human, machine, environment, and management—a comprehensive framework was developed to define system nodes, select relevant parameters, and apply guide words to identify potential hazards in a standardized manner. This approach allows for dynamic adjustment of influencing factors according to different application scenarios. The model was validated through a case study involving a restaurant, where it identified 65 potential gas-related hazards, over eight times more than traditional inspection approaches. Results demonstrate significant improvements in comprehensiveness, accuracy, and efficiency of hazard identification. This model provides a practical tool for urban gas safety management, supporting standardized hazard screening across various sectors including restaurants, residential buildings, schools, and commercial complexes. Furthermore, it lays the foundation for future integration of quantitative risk assessment and artificial intelligence-driven risk analysis, contributing to the digitalization and standardization of urban gas safety governance.

## 1. Introduction

In recent years, with the rapid advancement of urbanization in China, the urban gas industry has also experienced significant growth [1]. According to the latest data from the *Statistical Yearbook of Urban Construction in 2023*, published by the Ministry of Housing and Urban-Rural Development, key indicators such as gas user population, total gas supply, and gas penetration rate have shown continuous and substantial

**Data availability statement:** All data underlying the findings of this study are fully available within the manuscript. As this is a methodological study based on a structured hazard identification model (HAZOP-based), no external data collection, experimental measurements, or computational simulations were conducted. The complete minimal data set—including system nodes, influencing factors, parameters, guide words, and identified hazards—is fully described in the main text, particularly in the case application section and Table 2. No additional raw data were generated, and therefore no supplementary data files or repository deposition is required.

**Funding:** This study was supported by Jinhua Science and Technology Planning Project in the form of a grant awarded to B.T. (2025-4-283); the General Scientific Research Program of Zhejiang Ministry of Education in the form of a grant awarded to B.T. (Y202351241); the Zhejiang Education Science Planning Project in the form of a grant awarded to B.T. (2025SCG424); and the Scientific Research Planning Project of Yiwu City in the form of a grant awarded to B.T. (23-3-82). The specific roles of this author are articulated in the 'author contributions' section. The funders had no role in study design, data collection and analysis, decision to publish, or preparation of the manuscript.

**Competing interests:** The authors have declared that no competing interests exist.

increases [2]. Compared with 1993, the number of urban gas users increased from 94.39 million to 558.33 million in 2023, representing an approximate increase of 490%. The total gas supply rose from 20.79 billion cubic meters to 188.47 billion cubic meters—an increase of nearly 800%. Meanwhile, the gas penetration rate climbed from 27.90% to 98.25%, indicating that natural gas has become a dominant component of the urban energy structure. However, alongside this rapid development, frequent gas-related accidents have increasingly emerged, posing serious threats to public safety and social stability. According to incomplete statistics, a total of 612 gas accidents occurred nationwide in 2023, resulting in 77 fatalities and 434 injuries [3]. Among these accidents, one particularly severe accident—the LPG explosion at Fuyang BBQ Restaurant in Yinchuan City, Ningxia Province on June 21—caused 31 deaths and 7 injuries, triggering widespread concern and creating a major public safety crisis [4]. These events highlight the urgent need for enhanced risk prevention and control measures in urban gas systems.

Gas accidents are often triggered by undetected or uncontrolled potential hazards. If these risks can be comprehensively and accurately identified before an incident occurs, it may be possible to prevent the accident effectively [5]. Therefore, the identification and screening of accident hazards play a foundational role in accident prevention. Currently, urban gas hazard inspections primarily rely on manual patrols and random sampling. The effectiveness of these methods is largely constrained by the expertise and technical proficiency of personnel. Moreover, traditional paper-based documentation processes are not only cumbersome but also prone to information loss or delay, failing to meet the demands of modern urban gas safety management, which requires greater efficiency and accuracy [6]. Thus, developing a scientific, systematic, and operable method for identifying urban gas accident hazards holds great practical significance.

Urban gas accidents, particularly explosions and leaks, have increasingly drawn attention due to their potential for severe consequences. Numerous studies have explored the characteristics, causal factors, risk assessment methods, and safety management strategies associated with such accidents. Research on accident characteristics has revealed distinct spatiotemporal patterns, with higher incidence rates in residential and commercial areas, particularly during peak usage hours [7–9]. Investigations into accident causation have identified human error, equipment failure, and inadequate safety management as dominant contributing factors [10–13], often interacting in complex ways that amplify risk. Recent methodological advancements include the application of Bayesian networks, fuzzy AHP, fault tree analysis, and numerical simulations to enhance the accuracy and depth of risk assessment [14–17]. In terms of safety management, studies have proposed a range of strategies, from improving safety culture and operational procedures to the adoption of systematic hazard identification tools such as HAZOP [18–21].

Although several structured risk assessment methods—such as Failure Mode and Effects Analysis (FMEA), bow-tie analysis, and Fault Tree Analysis (FTA)—have been applied in safety engineering, each has limitations that constrain their effectiveness in complex urban gas systems [22]. FMEA primarily focuses on equipment failure

modes and their consequences, but often neglects systemic issues related to human behavior, organizational processes, and environmental conditions. Bow-tie analysis and FTA, while effective in visualizing causal pathways and mitigation barriers, typically requires prior knowledge of initiating events and is less suited for proactively identifying unknown or unanticipated hazards. These approaches may therefore fall short in contexts where risks emerge from dynamic interactions across technical, human, and managerial domains. In contrast, Hazard and Operability Analysis (HAZOP) offers a more systematic and structured framework for hazard identification. By systematically applying guide words (e.g., "no," "more," "less," "reverse") to predefined parameters across multiple dimensions—such as human, machine, environment, and management—HAZOP enables a comprehensive examination of potential deviations, even in the absence of historical incident data. This deviation-driven approach supports proactive, thorough, and scenario-independent hazard detection, making it particularly well-suited for socio-technical systems like urban gas networks.

Despite the richness of existing research on accident causation and risk assessment in urban gas systems, studies focusing on the theoretical framework of hazard identification remain relatively limited. In particular, there is a lack of standardized, systematic modeling approaches tailored specifically for urban gas systems. To address this gap, this study proposes a hazard identification model for urban gas systems based on the HAZOP deviation theory framework. The model integrates system node division, influencing factor identification, and parameter definition to establish a comprehensive, logically structured, and practically applicable methodology for identifying urban gas accident hazards. This research not only contributes to filling the theoretical gap in current hazard identification studies but also provides a new technical pathway and decision-support tool for urban gas safety management, offering significant value in promoting the intrinsic safety of urban gas systems.

## 2. Construction of the hazard identification model

To achieve systematic identification and management of urban gas accident hazards, this study constructs urban gas accident occurrence model and influencing factor system based on the mechanisms of such accidents. Building upon this foundation and integrating HAZOP deviation theory, a systematic hazard identification method tailored for urban gas systems is proposed [23]. Furthermore, the specific implementation steps for hazard identification are clearly defined. The resulting model enhances the scientific rigor and systematization of gas hazard recognition, providing theoretical support for subsequent risk assessment and the formulation of preventive measures. Fig 1 presents the overall framework of the proposed urban gas accident hazard identification model.

### 2.1. Types and mechanisms of urban gas accidents

Urban gas accidents can be broadly categorized into three types: fire, explosion, and poisoning/asphyxiation. These represent the primary forms of urban gas safety accidents, each governed by distinct occurrence mechanisms:

- **Fire accidents** refer to uncontrolled combustion caused by gas leakage or the ignition of normally burning gas by surrounding flammable materials, leading to casualties or property damage. The mechanism of fire accidents is illustrated in Fig 2.

- **Explosion accidents** occur when leaked gas accumulates within the explosive limit range in air and is ignited by an ignition source (e.g., open flame, static spark), triggering a rapid chemical reaction that generates high-temperature and high-pressure gases, causing structural and personnel damage. The process and key influencing factors are shown in Fig 3.

- **Poisoning or asphyxiation accidents** typically occur during incomplete combustion of gas, especially under poor ventilation conditions, where large amounts of toxic gases such as carbon monoxide are produced. Inhalation of these gases may result in poisoning or asphyxiation. The corresponding accident model is presented in Fig 4.

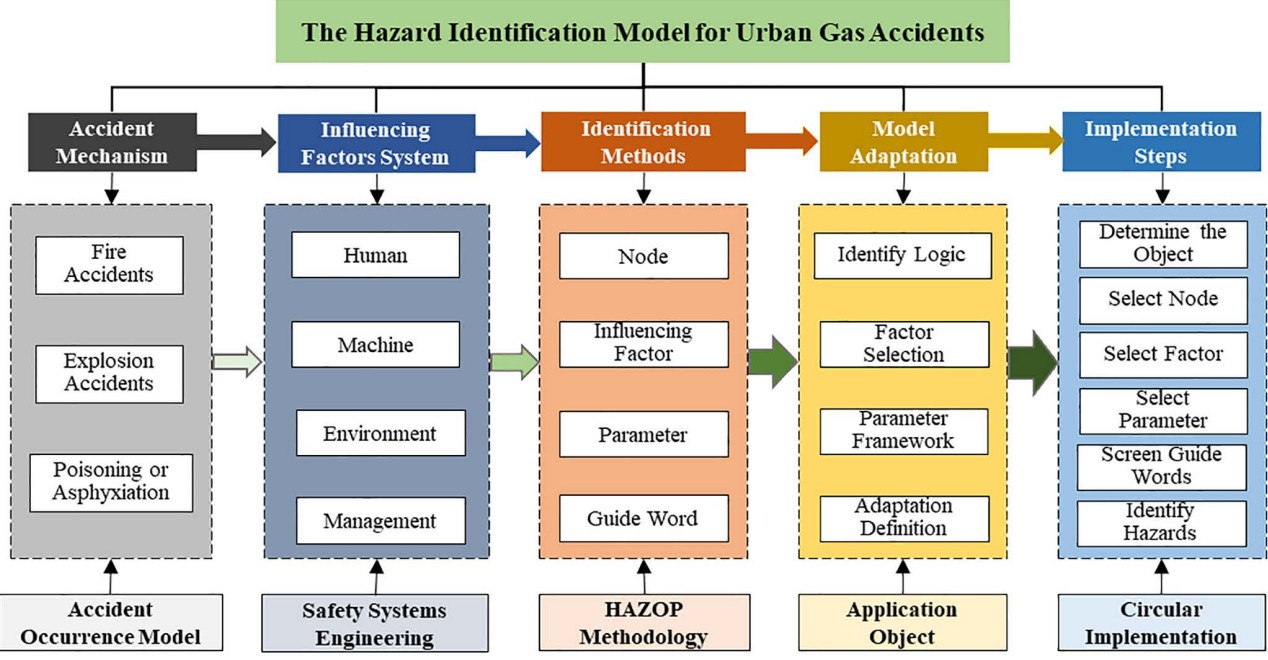

**Fig 1. Framework of the Urban Gas Accident Hazard Identification Model.**

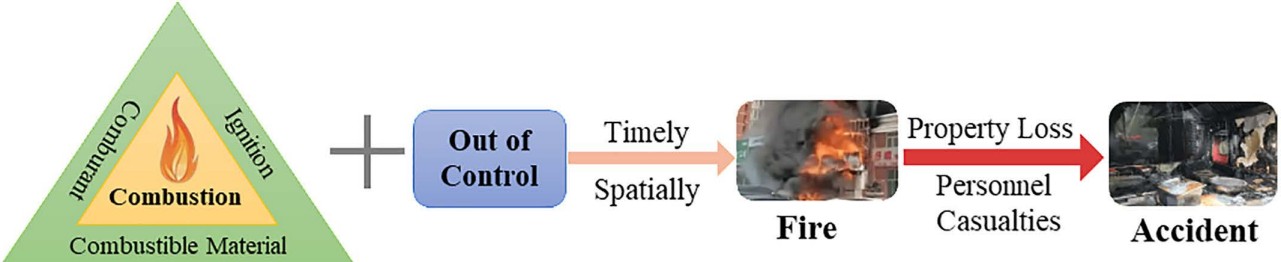

**Fig 2. Fire Accident Mechanism in Urban Gas Systems.**

## 2.2. Analysis of accident influencing factors

Based on systems engineering theory, the influencing factor system for urban gas accidents is constructed from the four dimensions: human, machine, environment, and management. This structured approach facilitates a comprehensive understanding of accident causation and provides a basis for systematic hazard identification. Taking into account the operational characteristics of urban gas systems, the influencing factor system is established and presented in Fig 5.

## 2.3. Hazard identification methods

Building upon the urban gas accident influencing factor system and incorporating HAZOP deviation analysis theory ('guide word + parameter = deviation') [24], a systematic urban gas accident hazard identification method is established, as illustrated in Fig 6.

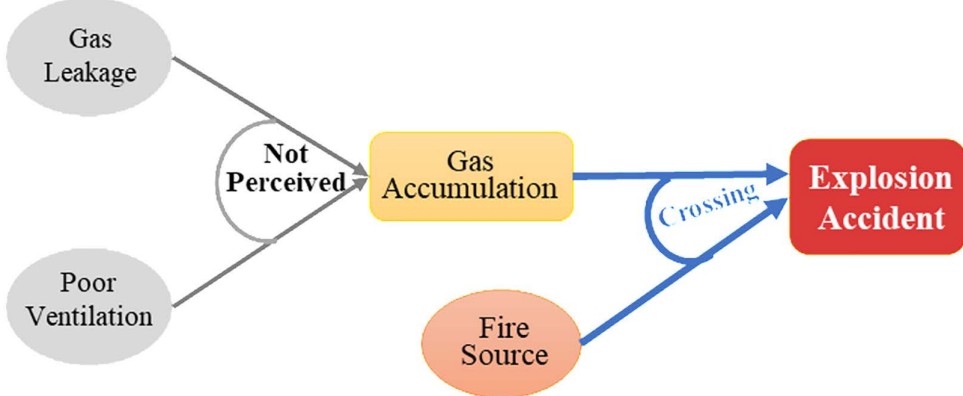

**Fig 3. Explosion Accident Mechanism in Urban Gas Systems.**

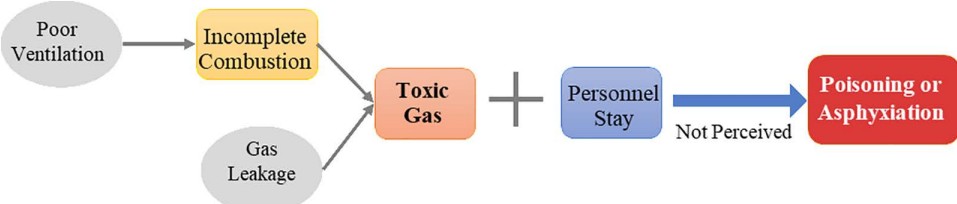

**Fig 4. Poisoning/Asphyxiation Accident Mechanism in Urban Gas Systems.**

The model integrates core elements such as system node division, parameter selection, guide word application, and deviation judgment, aiming to achieve comprehensive identification and evaluation of potential hazards within urban gas systems through a structured process.

## 2.4. Model adaptation and parameter definition

Inspired by the HAZOP methodology's fundamental concept, key parameters for the hazard identification model were redefined and adapted to suit the context of urban gas systems. The main definitions are summarized in Table 1.

It should be noted that the applicable parameters may vary across different analysis objects under the same influencing factor. Therefore, dynamic adjustment and screening are required based on the actual application context.

## 2.5. Implementation steps

Based on the constructed hazard identification model, this study proposes a complete set of implementation steps for urban gas accident hazard screening, which include the following eight steps:

### Step 1: Determine objectives and collect data

Select specific gas usage entities, such as residential households or restaurants, and gather relevant information including gas usage details, user profiles, regulations, equipment configurations, etc. It is recommended to combine field visits with on-site inspections to improve data accuracy and completeness.

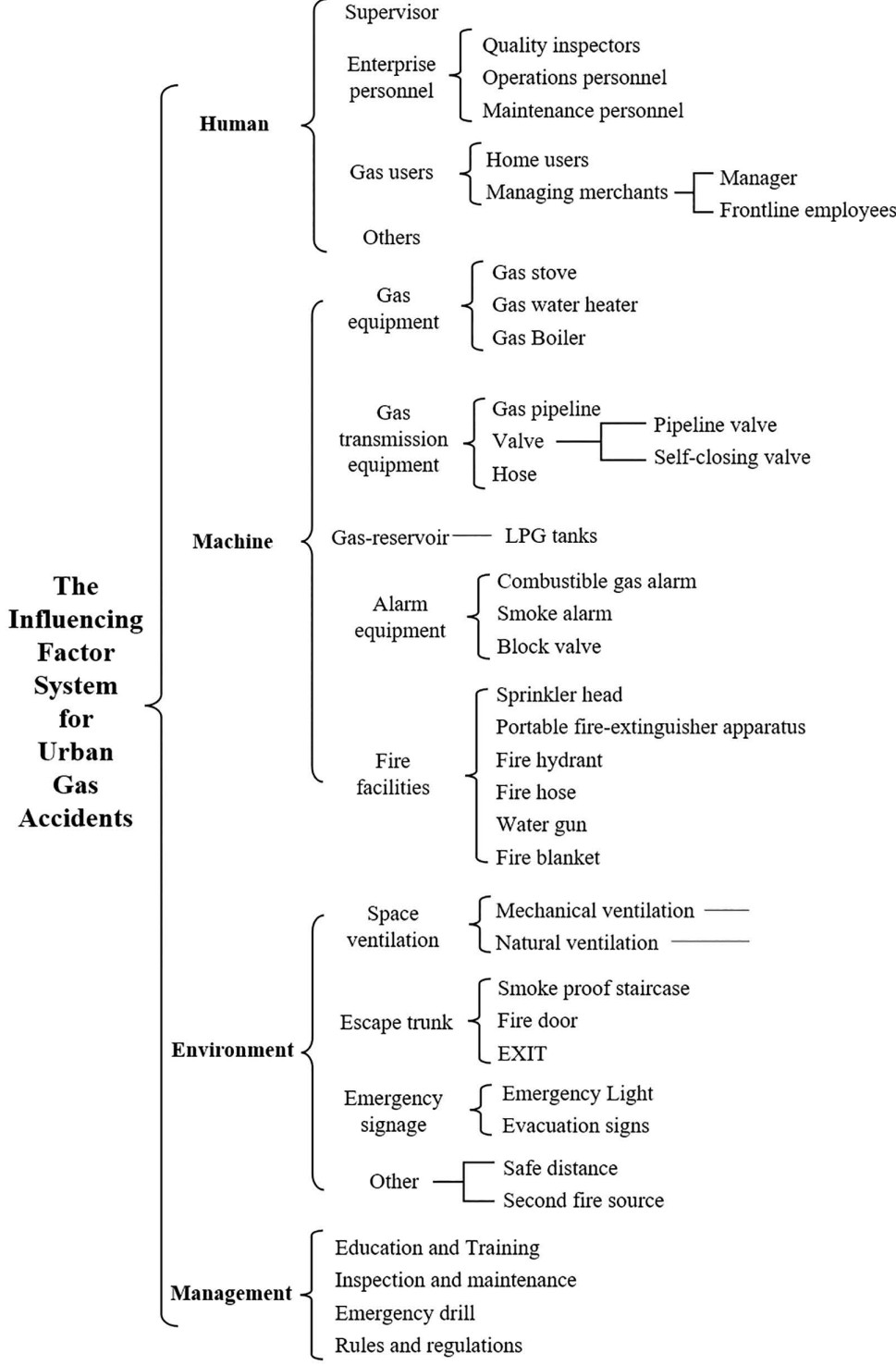

**Fig 5. Influencing Factor System for Urban Gas Accidents.**

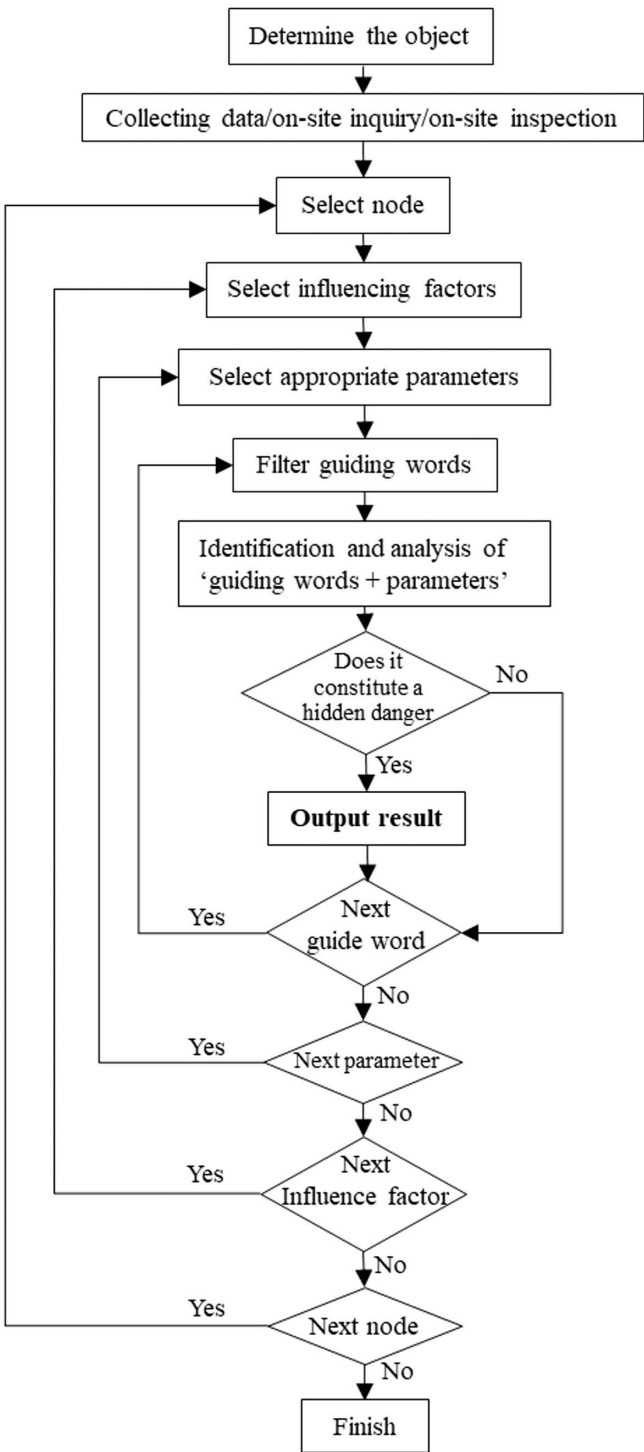

**Fig 6. Urban Gas Accident Hazard identification Method Diagram.**

**Table 1. Terminologies and Definitions Used in the Model.**

| Term | Definition | Description |
|------|-----------|-------------|
| **Hazard** | A condition with the potential to cause an accident, Corresponds to "deviation" in HAZOP. | Behaviors or states that do not comply with safety regulations or standards. |
| **Node** | An independent unit with clear boundaries, which represents basic units of analysis object. | Includes: human, machine, environment, management. |
| **Influencing Factor** | Variables affecting urban gas accidents. | Refers to end-level indicators in the influencing factor system shown in Fig 5. |
| **Parameter** | Specific indicators describing influencing factors | Includes: Human-related: awareness, knowledge, behavior, skill; Equipment/ environment-related: quantity, status, functionality; Management-related: establishment, implementation, applicability, effectiveness. |
| **Guide Word** | Keywords used to guide the evaluation of whether parameters pose a hazard | Includes: No (absence/unavailability), More (excess), Less (insufficiency), As well as (coexistence), Part of (partial), Reverse (opposite), Other than (deviation). |

## Step 2: Select nodes

Sequentially choose nodes from the "human-machine-environment-management" categories as the basic analysis units for hazard screening.

## Step 3: Select influencing factors

Identify relevant urban gas accident influencing factors for each node, ensuring comprehensive coverage and targeted relevance.

## Step 4: Select parameters

Refer to the established influencing factor system to clarify the category of selected influencing factors and sequentially select corresponding parameter indicators.

## Step 5: Screen guide words

Match appropriate guide words for each parameter to determine if there is a risk of deviation from safe conditions.

## Step 6: Identify hazards

Utilize the combination of "guide word + parameter" to identify for potential hazards. If hazards exist, record and output hazard descriptions.

## Step 7: Loop analysis

Determine if there are any unanalyzed guide words, parameters, influencing factors, or nodes. If so, return to Step 2 and continue the analysis until all applicable components have been thoroughly examined.

## Step 8: Organize and output results

Classify and organize all identified hazards, consolidate duplicates or similar hazards, eliminate ineffective or irrelevant issues, and ultimately produce a clear and detailed list of hazards for use in subsequent risk assessments and corrective action planning.

## 3. Case application

### 3.1. Overview of the case object

This study selected Restaurant H, located in Hangzhou, as the object for applying and validating the proposed hazard identification model. Restaurant H is a two-story building with a total area of approximately 160 square meters, of which the kitchen occupies about 30 square meters. The restaurant employs seven staff members: three kitchen workers, three front-of-house service personnel, and one manager. The main menu includes hot pot, barbecue, clay pot dishes, and stir-fried food. In daily operations, natural gas from pipelines is used for cooking in the kitchen, while electric stoves are used to heat hot pots at dining tables, and alcohol burners are used for dry pot dishes. Fig 7 presents the exterior view and kitchen layout of Restaurant H.

### 3.2. Hazard identification

Following the implementation steps outlined in Section 2.5, a comprehensive hazard identification process was conducted for Restaurant H. Based on the actual operational conditions of the restaurant and relevant regulatory requirements, 26 influencing factors were selected across the four system nodes — human, machine, environment, and management. Parameters related to each factor were defined, and appropriate guide words were applied. A total of 65 potential hazards were identified. Some representative results are summarized in Table 2 below.

### 3.3. Analysis of results

To gain deeper insights into the current status of hazard screening and management practices at Restaurant H, this study reviewed past safety inspection records and conducted interviews with the restaurant owner, staff members, and safety inspectors. The investigation revealed that Restaurant H's gas hazard screenings primarily relied on annual inspections by gas company personnel and spot checks by market supervision department officials. Traditional inspection methods typically involved checklists and manual patrols, identifying eight issues such as expired hoses, non-compliant hose lengths, objects hanging from pipelines, and unpowered combustible gas leak detectors.

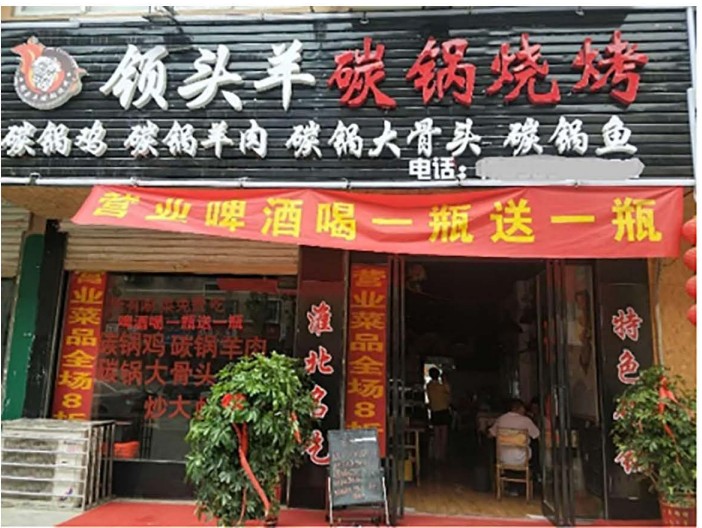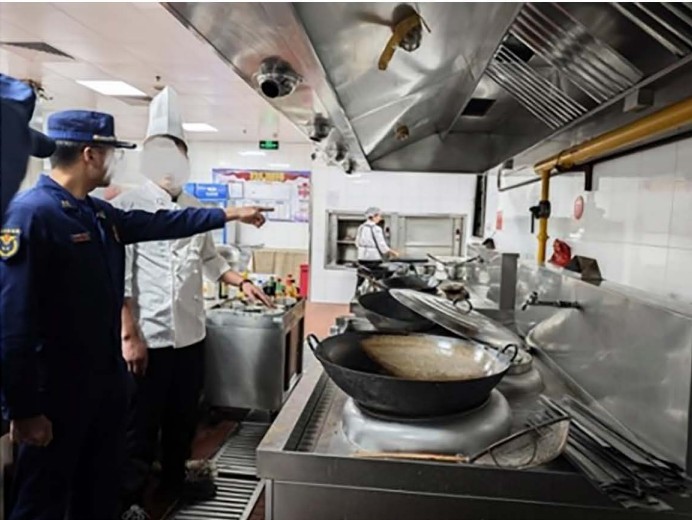

**Fig 7. Exterior View and Kitchen Layout of Restaurant H.**

**Table 2. Identified Gas Hazards in Restaurant H (Partial Results).**

| No. | Influencing Factor | Parameter | Guide Word | Hazard |
|---|---|---|---|---|
| 1 | Manager | Awareness | Less | Insufficient safety awareness – fails to check gas shut-off after daily operations. |
| 2 | | Knowledge | Part of | Inadequate gas safety knowledge – unaware of equipment service life. |
| 3 | | Skill | No | Lack of emergency response skills for gas leakage. |
| 4 | Frontline Staff | Awareness | Less | Insufficient safety awareness – fails to close the main gas valve after daily operations. |
| 5 | | Knowledge | Part of | Inadequate gas safety knowledge – unaware of accident-triggering conditions. |
| 6 | | Behavior | Reverse | Improper gas usage behavior – fire after gas release. |
| 7 | | Skill | Less | Insufficient emergency handling skills for gas leakage. |
| 8 | Gas Stove | Condition | Part of | Some stoves fail to ignite. |
| 9 | | Other | No | Gas stove lacks flame failure protection. |
| 10 | Pipeline | Condition | As well as | Items hanging on gas pipes. |
| 11 | | Condition | Other than | Pipeline suspended or under stress. |
| 12 | | Other | Other than | Loose pipe supports. |
| 13 | Hose | Quantity | More | Hose length exceeds 2 meters. |
| 14 | | Condition | As well as | Hose shows signs of aging. |
| 15 | | Other | More | Hose used beyond recommended lifespan. |

In comparison, the systematic hazard identification approach proposed in this study detected more than eight times the number of hazards. By analyzing 26 influencing factors across the dimensions of human, machine, environment, and management, and through dynamic adjustment of influencing factors, parameters, and guide words, the method significantly enhanced the relevance and comprehensiveness of the hazard identification outcomes.

The structured identification process ensured thorough and detailed examination, enabling systematic and holistic detection of hazards. The standardized model and procedural framework clarified the direction of hazard screening and improved the consistency and accuracy of the results. Therefore, the proposed systematic hazard identification method demonstrates strong practical applicability and potential for broader implementation in urban gas safety management.

## 4. Discussion

### 4.1. Advantages of the model

(1) Theoretical Innovation and Filling Research Gaps

This study pioneers the application of HAZOP deviation analysis methodology to the field of urban gas accident hazard identification, establishing a systematic and logical theoretical framework. Traditional research has primarily focused on causative analysis or risk assessment, with relatively limited attention given to methodological frameworks for hazard identification. By integrating HAZOP principles, this study extends its applicability beyond process systems, providing new theoretical support and methodological pathways for urban gas safety management.

(2) Clear Structure with High Practicality and Operability

The model employs a four-dimensional classification system encompassing human, machine, environment, and management dimensions, coupled with node division, parameter setting, and guide word selection mechanisms, resulting in a standardized and procedural hazard identification process. This approach is characterized by clear logic and explicit steps, facilitating rapid adoption and practical application by frontline safety managers or technicians. In the case of Restaurant H, the model successfully identified 65 potential hazards, more than eight times the number detected using traditional methods, demonstrating its efficiency and feasibility in real-world operations.

(3) High Flexibility with Good Adaptability and Scalability

The model design accounts for variability across different usage scenarios, allowing for dynamic adjustments of influencing factors, parameter settings, and guide word combinations based on specific objects. Whether applied in restaurants, residential buildings, school canteens, or commercial complexes, the model can be customized to suit diverse gas-using entities. This flexibility not only supports current applications but also lays the groundwork for broader dissemination in future research.

(4) Promotes Standardization and Normalization in Gas Safety Management

By establishing a unified hazard identification process and standard terminology system, the model enhances the standardization and consistency of gas safety management practices. Traditional inspection methods relying heavily on subjective experience often suffer from high omission rates. In contrast, the structured inspection process and standardized operational procedures of this model significantly improve the scientific rigor and accuracy of hazard identification, driving the transition from experience-based to systematic and standardized gas safety management.

## 4.2. Limitations of the model

(1) Predominantly Qualitative Analysis

The current model predominantly relies on qualitative analysis without establishing a quantitative evaluation index system. While qualitative analysis aids initial hazard recognition, incorporating quantitative assessments could enhance the precision and reliability of hazard identification in practical applications [25]. Future research should focus on introducing quantitative evaluation methods to complement and refine the existing qualitative framework.

(2) Limited Case Selection

The model was validated using a single case study in a restaurant setting, which, while representative, may limit the generalizability and robustness of the findings across other urban gas usage scenarios—such as residential buildings, schools, or industrial facilities. To better evaluate the model's applicability and stability, future work should include multi-case comparative analyses conducted across diverse environments and ideally by independent teams. Such efforts would help minimize potential bias and strengthen the empirical foundation of the proposed framework [26,27].

## 4.3. Directions for Future Research

(1) Expanding Application Scenarios

While the model has been demonstrated to be effective in restaurant settings, its potential extends beyond this domain. Future research could explore its application in other types of gas-using entities, such as residential homes, school cafeterias, and commercial complexes. Continuous optimization and adjustment of model parameters will enable adaptation to more diverse scenarios, thereby validating its broad applicability.

(2) Integration with Other Safety Management Systems

Exploring the integration of this model with other safety management systems (e.g., ISO 45001, GB/T 28001) could promote the standardization and intelligent development of urban gas safety management. By consolidating various safety management systems, a more comprehensive and systematic safety management strategy can be developed, enhancing the overall level of urban gas safety management.

   

## 5. Conclusion

This study addresses the issue of urban gas accident hazard identification by proposing a hazard identification model based on HAZOP deviation analysis. Through the construction of an "human-machine-environment-management" influencing factor system, combined with parameter setting and guide word selection mechanisms, a systematic and structured hazard identification method was developed and validated through a practical case study. The findings indicate that the model not only exhibits strong theoretical innovation but also demonstrates good practicality and operability at the implementation level. However, there is still room for improvement, particularly in terms of quantitative evaluation and multi-case validation. Future research should continue to explore the application of new technologies, expand the scope of model application, and contribute to the refinement and standardization of urban gas safety management practices.

## Author contributions

**Conceptualization:** Bin Tian.

**Data curation:** Bin Tian, Haibing Li.

**Formal analysis:** Bin Tian, Zibo Zhou.

**Funding acquisition:** Bin Tian, Xiaojun Cui, Zibo Zhou, Wei Shi.

**Investigation:** Zhihong Hu, Zibo Zhou, Wei Shi.

**Methodology:** Bin Tian, Haibing Li, Zhihong Hu.

**Project administration:** Bin Tian.

**Resources:** Xiaojun Cui, Zhihong Hu, Zibo Zhou, Wei Shi.

**Software:** Bin Tian.

**Supervision:** Bin Tian, Zhihong Hu.

**Validation:** Bin Tian, Zibo Zhou.

**Writing – original draft:** Bin Tian, Haibing Li, Xiaojun Cui, Zhihong Hu.

**Writing – review & editing:** Bin Tian, Haibing Li, Zhihong Hu, Wei Shi.

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
