## [Decision Letter · Decision Letter 0]

24 Aug 2025

Dear Dr. Tian,

Thank you for submitting your manuscript to PLOS ONE. After careful consideration, we feel that it has merit but does not fully meet PLOS ONE’s publication criteria as it currently stands. Therefore, we invite you to submit a revised version of the manuscript that addresses the points raised during the review process.

We look forward to receiving your revised manuscript.

Kind regards,

Muhammad Athar, PhD

Academic Editor

PLOS ONE

Journal Requirements:

3. Please note that PLOS One has specific guidelines on code sharing for submissions in which author-generated code underpins the findings in the manuscript. In these cases, we expect all author-generated code to be made available without restrictions upon publication of the work. Please review our guidelines at https://journals.plos.org/plosone/s/materials-and-software-sharing#loc-sharing-code and ensure that your code is shared in a way that follows best practice and facilitates reproducibility and reuse.

This study was supported by the General Scientific Research Program of Zhejiang Ministry of Education (Y202351241); Zhejiang Education Science Planning Project (2025SCG424); Scientific Research Planning Project of Yiwu City (23-3-82).

6. Please note that funding information should not appear in any section or other areas of your manuscript. We will only publish funding information present in the Funding Statement section of the online submission form. Please remove any funding-related text from the manuscript.

7. We note you have included a table to which you do not refer in the text of your manuscript. Please ensure that you refer to Table 1 in your text; if accepted, production will need this reference to link the reader to the Table.

Reviewers' comments:

Reviewer's Responses to Questions

**Comments to the Author**

1. Is the manuscript technically sound, and do the data support the conclusions?

Reviewer #1: Yes

Reviewer #2: Yes

2. Has the statistical analysis been performed appropriately and rigorously?

Reviewer #1: N/A

Reviewer #2: Yes

3. Have the authors made all data underlying the findings in their manuscript fully available?

Reviewer #1: Yes

Reviewer #2: Yes

4. Is the manuscript presented in an intelligible fashion and written in standard English?

Reviewer #1: Yes

Reviewer #2: Yes

Reviewer #1: The manuscript presents an interesting application of HAZOP for urban gas accident risk analysis by integrating identified hazards with potential accident scenarios. The use of a case study to validate the framework is a commendable approach. However, the validation strategy could be further strengthened by involving subject matter experts to independently assess the proposed framework. Additionally, incorporating more case studies conducted by individuals or teams not involved in the development of the method would help minimize potential bias and enhance the credibility of the findings.

Reviewer #2: This manuscript addresses an important and timely issue of urban gas safety by proposing a hazard identification model based on HAZOP deviation theory. The integration of human, machine, environment, and management dimensions into a structured hazard identification framework is innovative and practically valuable. The case study demonstrates strong potential for real-world application. The paper is generally well-written, logically structured, and supported by relevant literature. However, there are some issues that need to be modified:

1. The introduction reviews urban gas accident statistics and prior methodologies but could better highlight why HAZOP, specifically, is superior compared to other structured methods (FMEA, bow-tie, etc.).

2. Figures and tables are informative, but caption styles are inconsistent (e.g., “Fig.” vs. “Figure”).

3. References are not arranged in order.

4. Overuse of capitalization in heading: "Construction of the Hazard Identification Model". It should not capitalize every word. Please revise this for other headings.

5. keywords are not in alphabetical order. please revise.

6. In introduction section, numbers are inconsistent. sometimes with two decimals (20.79 billion m³), three and four decimals (94.394 million to 558.3287 million) and sometimes none (31 deaths). Must standardize this.

**Do you want your identity to be public for this peer review?** For information about this choice, including consent withdrawal, please see our Privacy Policy

Reviewer #1: **Yes: ** Noor Diana Binti Abdul Majid

Reviewer #2: **Yes: ** Muhammad Arslan Jameel Malik

---

## [Author Response · Author response to Decision Letter 1]

28 Aug 2025

Dear Editor and reviewers

On behalf of all authors, we thank the Academic Editor and reviewers for their constructive comments and suggestions, which have helped us improve the quality and clarity of the manuscript. Below, we provide a point-by-point response to all comments. All changes in the manuscript are highlighted in the revised version with track changes.

Journal Requirements:

1. [PLOS ONE style]

Response: We have carefully reviewed the PLOS ONE formatting guidelines and reformatted the manuscript accordingly, including file naming, section headings, and reference style.

2. [ORCID]

Response: The corresponding author (Bin Tian) has verified his ORCID iD (https://orcid.org/0009-0001-3391-9158) in Editorial Manager.

3. [Code sharing]

Response: This study does not involve author-generated code. Therefore, this requirement does not apply.

4. [Data availability]

Response: All data underlying the findings of this study are fully available within the manuscript. As this is a methodological study based on a HAZOP-based hazard identification framework, no external data collection or computational simulations were conducted. The model application, including all relevant parameters, deviation analyses, and resulting hazard identifications, is fully described in the text. Therefore, the manuscript itself constitutes the complete minimal data set, and no additional data deposition is required.

5. [Role of funders]

Response: We have added the following statement in the cover letter: "The funders had no role in study design, data collection and analysis, decision to publish, or preparation of the manuscript."

6. [Funding in manuscript]

Response: All funding-related text has been removed from the manuscript. Funding information is now only included in the submission system.

7. [Table reference]

Response: We have added a reference to Table 1 in Section 2.4: " The main definitions are summarized below as Table 1."

8. [Citation recommendations]

Response: No specific citations were recommended by the reviewers. All references have been verified for accuracy and relevance.

9. [Reference list]

Response: The reference list has been checked and formatted according to PLOS ONE’s sequential numbering style. All citations are accurate, and no retracted papers are cited.

---

Reviewer #1:

Comment: The validation strategy could be further strengthened by involving subject matter experts and more external case studies to reduce bias.

Response: We fully agree with this insightful comment. In response, we have revised the subsection (4.2 Limitations of the Model), in which we explicitly acknowledge the limitation of single-case validation and emphasize the need for future multi-case and independent-team validation to enhance objectivity and generalizability. The following text has been added to Section 4.2(2):

"To better evaluate the model’s applicability and stability, future work should include multi-case comparative analyses conducted across diverse environments and ideally by independent teams. Such efforts would help minimize potential bias and strengthen the empirical foundation of the proposed framework [26,27]."

---

Reviewer #2:

Comment 1: Better highlight why HAZOP is superior compared to other methods (e.g., FMEA, bow-tie).

Response: We have added a comparative statement in the Introduction:

" Although several structured risk assessment methods—such as Failure Mode and Effects Analysis (FMEA), bow-tie analysis, and Fault Tree Analysis (FTA)—have been applied in safety engineering, each has limitations that constrain their effectiveness in complex urban gas systems [22]. FMEA primarily focuses on equipment failure modes and their consequences, but often neglects systemic issues related to human behavior, organizational processes, and environmental conditions. Bow-tie analysis and FTA, while effective in visualizing causal pathways and mitigation barriers, typically requires prior knowledge of initiating events and is less suited for proactively identifying unknown or unanticipated hazards. These approaches may therefore fall short in contexts where risks emerge from dynamic interactions across technical, human, and managerial domains. In contrast, Hazard and Operability Analysis (HAZOP) offers a more systematic and structured framework for hazard identification. By systematically applying guide words (e.g., “no,” “more,” “less,” “reverse”) to predefined parameters across multiple dimensions—such as human, machine, environment, and management—HAZOP enables a comprehensive examination of potential deviations, even in the absence of historical incident data. This deviation-driven approach supports proactive, thorough, and scenario-independent hazard detection, making it particularly well-suited for socio-technical systems like urban gas networks."

Comment 2: Inconsistent figure captions (e.g., “Fig.” vs “Figure”).

Response: All figure captions have been standardized to “Fig X.” format, in accordance with PLOS ONE guidelines.

Comment 3: References are not in order.

Response: The reference list has been re-ordered according to the sequence of citation in the text, following PLOS ONE’s numbering system.

Comment 4: Overuse of capitalization in headings.

Response: All section headings have been revised to sentence case (only first word capitalized), e.g., “Construction of the hazard identification model”.

Comment 5: Keywords not in alphabetical order.

Response: The keywords have been re-ordered alphabetically:

"Case study; Framework development; Hazard identification model; HAZOP; Urban gas accident"

Comment 6: Inconsistent number formatting in Introduction.

Response: All numerical values have been standardized:

“94.394 million” → “94.39 million”

“558.3287 million” → “558.33 million”

Other values (e.g., 27.90%, 20.79 billion m³) were already appropriately formatted.

---

We believe that the revised manuscript has addressed all concerns and is now suitable for publication in PLOS ONE. Thank you again for the opportunity to improve our work.

---

## [Decision Letter · Decision Letter 1]

15 Sep 2025

A HAZOP-based hazard identification model for urban gas accidents: Development and empirical validation

PONE-D-25-41119R1

Dear Dr. Tian,

We’re pleased to inform you that your manuscript has been judged scientifically suitable for publication and will be formally accepted for publication once it meets all outstanding technical requirements.

Kind regards,

Muhammad Athar, PhD

Academic Editor

PLOS ONE

Additional Editor Comments (optional):

Reviewer #1:

Reviewer #2:

Reviewers' comments:

Reviewer's Responses to Questions

**Comments to the Author**

Reviewer #1: All comments have been addressed

Reviewer #2: All comments have been addressed

2. Is the manuscript technically sound, and do the data support the conclusions?

Reviewer #1: Yes

Reviewer #2: Yes

3. Has the statistical analysis been performed appropriately and rigorously?

Reviewer #1: Yes

Reviewer #2: Yes

4. Have the authors made all data underlying the findings in their manuscript fully available?

Reviewer #1: Yes

Reviewer #2: Yes

5. Is the manuscript presented in an intelligible fashion and written in standard English?

Reviewer #1: Yes

Reviewer #2: Yes

Reviewer #1: The authors have carefully revised the manuscript according to the reviewers’ comments. All suggested changes and clarifications have been incorporated, ensuring that the issues raised have been thoroughly addressed. The language has been refined, additional explanations have been provided where necessary, and supporting references have been included to strengthen the arguments. Technical corrections and formatting improvements were also made to enhance readability. The revised version reflects the reviewers’ constructive feedback and demonstrates significant improvement in content, structure, and clarity. We believe the manuscript now meets the required standards for publication and respectfully resubmit for consideration.

Reviewer #2: (No Response)

**Do you want your identity to be public for this peer review?** For information about this choice, including consent withdrawal, please see our Privacy Policy

Reviewer #1: **Yes: ** Noor Diana Binti Abdul Majid

Reviewer #2: **Yes: ** Muhammad Arslan Jameel Malik

---

## [Editor Report · Acceptance letter]

PONE-D-25-41119R1

PLOS ONE

Dear Dr. Tian,

I'm pleased to inform you that your manuscript has been deemed suitable for publication in PLOS ONE. Congratulations! Your manuscript is now being handed over to our production team.

Kind regards,

on behalf of

Dr. Muhammad Athar

Academic Editor

PLOS ONE